# Study on the Adaptive Regulation of Light on the Stress Response of Mandarin Fish (*Siniperca chuatsi*) with Re-Feeding after Starvation

**DOI:** 10.3390/ani13162610

**Published:** 2023-08-13

**Authors:** Jian Zhou, Qiang Li, Zhipeng Huang, Lu Zhang, Chengyan Mou, Zhongmeng Zhao, Han Zhao, Jun Du, Xiaojun Yang, Xufang Liang, Yuanliang Duan

**Affiliations:** 1College of Fisheries, Huazhong Agricultural University, Wuhan 430070, China; zhoujian980@126.com; 2Fisheries Institute, Sichuan Academy of Agricultural Sciences, Chengdu 611731, China; liq7920@126.com (Q.L.); h3392078@163.com (Z.H.); zhanglu425@163.com (L.Z.); 15927383463@163.com (C.M.); 18227552594@163.com (Z.Z.); zhaohan232323@163.com (H.Z.); dujun9100@126.com (J.D.); 3Western Aquatic Seed Industry Co., Ltd., Mianyang 621000, China

**Keywords:** *Siniperca chuatsi*, light, re-feeding behaviors, starvation, stress response

## Abstract

**Simple Summary:**

Environmental factors have a significant impact on the feeding behavior, yield, and quality of *Siniperca chuatsi*. Previous studies of potential correlations between light and fish behaviors have mainly focused on phototaxis, changes to circadian rhythms, and growth and development. In the present study, at 11.15 ± 2.01 lx, *S. chuatsi* sustained relatively lower stress in response to re-feeding after starvation and digestive enzyme activities in the intestine were the highest, indicating that the light intensity was most suitable for re-feeding of *S. chuatsi* after starvation. Under light conditions, *S. chuatsi* only preys on fish, but it also preys on shrimp under dark conditions, due to weak light reducing the accuracy of hunting. Thus, under weak light, *S. chuatsi* is most likely to consume compound feed. Moreover, reducing the light intensity can increase the activity of digestive enzymes in the intestine while alleviating the stress response and facilitate successful domestication of *S. chuatsi*. Collectively, the results of the present study suggest that the suitable light intensity can accelerate the adaptation of *S. chuatsi* to stress caused by re-feeding.

**Abstract:**

Light influences the stress response to environmental stimuli and feeding behaviors of *Siniperca chuatsi* and, thus, is an important regulator of normal growth and development. In this study, we first explored the important role of light on the digestive and stress capacity of *S. chuatsi* by studying the changes in physiological and biochemical indicators of *S. chuatsi*, taking the re-feeding after starvation as the constant environmental stimulus and the light intensity as the adjustable environmental stimulus. The activity of protease and lipase was generally higher in the stomach tissues than in the intestinal tissues, especially lipase, which was higher in stomach tissues under all light conditions, and the protease and lipase activity peaked in the stomach tissues of *S. chuatsi* at a light intensity of 18.44 ± 3.00 lx and in intestinal tissues at 11.15 ± 2.01 lx, respectively, indicating that greater light intensity increased the digestive capacity of stomach tissues, whereas lower light intensity facilitated the digestive capacity of intestinal tissues. The tissues of the gill, stomach, and intestine had relatively high activity of stress-related enzymes, whereas the tissues of the brain, kidney, liver, and plasma samples had relatively low activity of enzymes. Collectively, the results show that light intensity at 11.15 ± 2.01 lx promoted digestive capacity in the intestine and enhanced the anti-stress ability of *S. chuatsi* in response to stress induced by re-feeding after starvation. These findings should prove useful for artificial breeding of *S. chuatsi*.

## 1. Introduction

The Mandarin fish (*Siniperca chuatsi*), a member of the order Perciformes and family Serranidae, is an economically important freshwater species in countries throughout East Asia, including China [1]. Environmental factors have significant impacts on the growth, yield, and quality of *S. chuatsi*, such as light intensity, temperature, and microorganisms [2,3,4]. Various fish species reduce or stop feeding in response to colder temperatures in the winter and start feeding again in the spring [5,6]. To avoid financial loss, most farmers choose to reduce the breeding time and sell *S*. *chuatsi* prior to maturation when the body weight reaches about 500 g [7]. The body weight of *S*. *chuatsi* in aquaculture is generally 2–2.5 kg and can reach more than 5 kg in natural waters [8]. Sexual maturity of male and female *S*. *chuatsi* occurs at the age of 1 and 2 years, respectively [9]. Hence, *S*. *chuatsi* cultured for less than 1 year has not yet reached the highest market value.

Previous studies of potential correlations between light and fish behaviors have mainly focused on phototaxis [10,11], changes to circadian rhythms [12], and growth and development [13,14,15]. Recent studies of *S. chuatsi* have mainly focused on selective breeding [16], breeding technologies [17], feed formulations [18], nutritional quality [19], and disease susceptibility [20]. A few studies have investigated the re-feeding behaviors and morphological changes of *S. chuatsi* after starvation, such as Lu et al. [21] for growth hormone receptors, Chen et al. [22] for insulin-like growth factor 1 receptors, Zhu et al. [23] for microRNA profiles, Wang et al. [24] for gene expression profiles, and Shi et al. [25] for memory function. However, no study has yet to investigate the influence of light on the feeding behaviors of *S*. *chuatsi* and only one assessed the impact of light on aquatic plant coverage, with a focus on prey consumption and growth of *S. chuatsi* [2]. The studies mentioned above confirmed that light with satisfactory intensity and color promotes the growth and survival of fish and that too much light can be hazardous via disruption of circadian rhythm. Therefore, it is particularly important to assess the impact of light on fish behavior.

When a fish is in a healthy state and undergoes a negative impact due to the impact of the external environment, changes in the molecular level of the body are an important indicator, such as digestive enzymes and stress-related enzymes. The activity of digestive enzymes (protease, lipase, amylase) can reflect the digestive ability of fish to food [26]. Whether the fish is under hunger stress or fed with different feeds, the activity of digestive enzymes can reflect the adaptability of the fish to the environment and the health status of the fish to a certain extent [27]. The activity of stress-related enzymes in the fish can directly reflect whether the fish is in a state of stress. Acid phosphatase (ACP), alkaline phosphatase (AKP), lactate dehydrogenase (LDH), superoxide dismutase (SOD), catalase (CAT), alanine aminotransferase (GPT), aspartate aminotransferase (GOT), and other stress-related enzymes are often used to measure the stress response and adaptability of fish to stress after being stimulated [28,29,30]. Reasonable attention to these indicators can provide very useful suggestions and guidance for our breeding activities.

An appropriate amount of light can alleviate stress on the fish. Re-feeding after starvation can cause a stress response in *S*. *chuatsi*. In the present study, different light intensities were applied to mimic natural periodic light–dark cycles on *S. chuatsi* to assess changes to the activity of enzymes involved in digestion and stress and to determine whether the satisfactory intensity of light can alleviate stress induced by re-feeding after starvation, as well as to clarify the underlying mechanisms. The results of this study provide theoretical guidance and references for artificial breeding of *S. chuatsi* to quickly adapt to stress caused by re-feeding after starvation.

## 2. Materials and Methods

### 2.1. Ethics Statement

The study protocol was approved by the Animal Ethics Committee of the Fisheries Institute of Sichuan Academy of Agricultural Sciences (Chengdu, China; approval no. 20170226001A) and conducted in accordance with the Guide for the Care and Use of Laboratory Animals.

### 2.2. Source of Experimental Organisms and Experimental Design

In total, 150 young *S. chuatsi* (mean length, 18.63 ± 0.80 cm; mean weight, 76.55 ± 7.87 g) were obtained from Western Aquatic Seed Industry Co., Ltd. (Mianyang, China) and acclimated to laboratory conditions for 7 days prior to experimentation. Temporary feeding conditions: (1) 7 m^3^ of water in the circulating aquarium, (2) continuously oxygenated for 24 h, and (3) fed daily at 8:00 and 20:00. The nutritional composition of the feed is shown in Appendix A. Moreover, 150 fish were randomly assigned to 15 tanks (*n* = 10/tank, 80 × 60 × 55 cm^3^). The fish were cultured in an automatic circulating water system under artificially reduced natural lighting conditions. The light intensity of this study was based on the experience of farmers, the surface water light intensity in the commercial fish aquaculture water was about 30 lx, and in the water of seedling domestication was about 10 lx. The experimental system consists of three identical internal devices named S1, S2, and S3 (Figure 1). S1, S2, and S3 are parallel repeating relationships, with an interval of 1 m between S1 and S2 and the same distance between S2 and S3. The distance between S1 and the wall with light source is 0.5 m, and the same applies to S2 and S3. Such a device can ensure that the source and variation of the experimental light source in this study tend to be consistent. Except for the light changing with the natural photoperiod, other factors (temperature, dissolved oxygen, sound, etc.) in this experimental system were basically consistent. The light intensity was measured at the center of the bottom of the aquarium with a ZDS-10W underwater illuminometer (TIANWEI, China). The measurement time points were 8:30, 13:30, and 17:30 under static conditions without gassing with air, and the mean light intensities of groups 1–5 were 21.04 ± 2.72 lx, 18.44 ± 3.00 lx, 15.75 ± 2.88 lx, 13.43 ± 2.35 lx, and 11.15 ± 2.01 lx, respectively. More detailed environmental data (light intensity, temperature, and dissolved oxygen) are shown in Appendix A. Feed was withheld for 7 days prior to re-feeding, and feeding with satiety (3% of body weight) at 8: 00 a.m. was carried out on the 8th day. Fish samples were randomly collected two hours later; 2 fish were collected from each tank in S1 and S2, respectively; and 1 fish was collected from each tank in S3. That is, a total of 5 fish were collected from each treatment group. No fish died during the experimental period. Moreover, the stress-related indicators and digestive enzymes of *Siniperca chuatsi* before hunger are shown in Appendix A.

### 2.3. Tissue Sample Collection

The fish were euthanized with a high dose (500 mg/L) of tricaine methanesulfonate (MS-222; Sigma-Aldrich Chemie GmbH, Schnelldorf, Germany) [31] prior to dissection for the collection of gill, brain, intestine, stomach, kidney, and liver tissues in addition to plasma samples. Before dissection, the fish were disinfected with 75% alcohol. When collecting intestinal tissues, the food had already reached the middle and posterior segments of the intestine. By gently pressing, food was squeezed out of the intestine and intestinal tissue was collected. By gently pressing, food was squeezed out of the intestine, and the intestinal tissue was collected after being washed with phosphate-buffered saline (PBS saline, pH 7.4) [32]. Plasma samples were collected through the tail vein [33]. The tissue samples were homogenized in 0.9% normal saline (1:9 *w/v*) in an ice water bath using a homogenizer, then centrifuged at 3500 rpm for 10 min, and the supernatant was collected for analysis of enzyme activity. The plasma samples were left overnight at 4 °C, then centrifuged at 3500 rpm for 10 min, and the supernatant was collected for analysis of enzyme activity.

### 2.4. Enzymatic Activity Analysis

In this study, the enzymatic activities of pepsin, lipase, and *α*-amylase in the intestine and stomach were detected using a Pepsin Assay Kit, a Lipase Assay Kit, and an α-Amylase Assay Kit (Nanjing Jiancheng, Bioengineering Institute, China). The enzymatic activities of ACP, AKP, LDH, SOD, CAT, GPT, and GOT in the gill, brain, intestine, stomach, kidney, liver, and plasma were detected using an Acid Phosphatase Assay Kit, an Alkaline Phosphatase Assay Kit, a Lactate Dehydrogenase Assay Kit, a Superoxide Dismutase Assay Kit, a Catalase (CAT) Assay Kit, an Alanine Aminotransferase Assay Kit, and an Aspartate Aminotransferase Assay Kit (Nanjing Jiancheng Bioengineering Institute, China). Total protein was determined with a Total Protein Assay Kit (Nanjing Jiancheng Bioengineering Institute, China). The operation steps were carried out according to the operation guide of the reagent kit, and the operation guide of the corresponding reagent kit can be searched for on the website (http://www.njjcbio.com/, accessed on 8 June 2023). In addition, the calculation formulas for these enzyme activities are detailed in the Supplemental Material.

### 2.5. Data Analysis

Statistical analyses were performed with Excel 2016 (Microsoft Corporation, Redmond, WA, USA) and IBM SPSS Statistics for Windows (version 19.0.; IBM Corporation, Armonk, NY, USA). Before formal analysis, the data were subjected to normality testing and equality of variance testing using SPSS 19.0 software. Under the same tissues, the enzyme activity under different light intensities fully or approximately conformed to the normal distribution. Therefore, this study was based on a one-way analysis of variance of enzyme activity under the same tissue and different light intensities. One-way analysis of variance was used to identify statistically significant differences among groups, and multiple comparisons were performed using Duncan’s range test. The results are presented as the mean ± standard deviation. A probability (*p*) value ≤ 0.05 was considered statistically significant.

## 3. Results

### 3.1. Total Protein Content Analysis

The total protein content of all tissues and plasma samples is shown in Figure 2. The maximum protein content was observed in most samples at a lower light intensity of 11.15 ± 2.01 lx, except for the brain and liver, for which the total protein content was significantly higher than that under one or more other light intensities. As the light intensity weakened, the variation pattern of protein content varied among different tissues, but when the light intensity decreased to 11.15 ± 2.01 lx, it basically showed an increasing trend, which meant that a lower light intensity increased the protein content in *S. chuatsi*. The total protein content in plasma samples (average value, 23,348.48 ± 3770.07 μg mL^−1^) was much higher than in gill, brain, intestine, stomach, kidney, and liver tissue samples (average value, 3288.18–8157.15 μg mL^−1^).

### 3.2. Digestive Enzyme Activity Analysis

The activity of the digestive enzymes is shown in Figure 3. There was no significant change in protease activity in the intestinal tissues under different light intensities, but it tended to increase with decreasing light intensity (11.15–18.44 lx). Protease activity in the stomach tissues peaked at 18.44 ± 3.00 lx with 2.39 ± 0.78 U ml^−1^. Lipase activity in the intestinal tissues tended to increase with decreasing light intensity (11.15–18.44 lx, *p* < 0.01). In the stomach tissues, lipase activity peaked at 18.44 ± 3.00 lx with 3341.90 ± 23.35 U g^−1^ prot and was significantly higher at 18.44–15.75 lx than at other light intensities (*p* < 0.01). The activity of *α*-amylase in the intestinal and stomach tissues, except at 21.04 ± 2.72 lx, showed basically the same rate of change and peaked at 13.43 ± 2.35 lx prior to decreasing again. The activity of protease and lipase was generally higher in the stomach tissues than in the intestinal tissues, especially lipase, which was higher in stomach tissues under all light conditions.

### 3.3. Stress-Related Enzyme Activity Analysis

As shown in Figure 4, the changes in gill, brain, and stomach tissues were basically consistent, showing a trend of first increasing, then decreasing, and finally increasing. The changes in the intestine and kidney tissues were basically consistent, showing a trend of first decreasing, then increasing, and finally decreasing. The plasma samples showed a trend of first decreasing, then increasing, then decreasing, and finally increasing. The liver tissues showed a trend of first decreasing and then increasing. Except for plasma samples, the trend of changes in ACP activities in other tissues reached significant levels (*p* < 0.01 or *p* < 0.05). At a light intensity greater than 15.75 ± 2.88 lx, ACP activity was more likely to be lower in all samples. At a light intensity lower than 15.75 ± 2.88 lx, ACP activity was more likely to peak in all samples. ACP activity in intestinal tissues was very high, with an average of 42.95 ± 5.75 King unit g^−1^ prot and a maximum of 48.76 ± 5.08 King unit g^−1^ prot. In addition, ACP activity was the lowest in gill tissues, with an average of 7.20 ± 3.38 King unit g^−1^ prot.

As shown in Figure 5, the changes in the brain and stomach tissues were basically consistent, showing a trend of first increasing, then decreasing, and finally increasing. The changes in the intestine, kidney, and liver tissues were basically consistent, showing a trend of first decreasing, then increasing, and finally decreasing. The gill tissues showed a trend of first increasing and then decreasing. The plasma samples showed a trend of first decreasing and then increasing. Except for kidney tissues, the trend of changes in AKP activity in other tissues reached significant levels (*p* < 0.01 or *p* < 0.05). When the light intensity was less than 15.75 ± 2.88 lx, most tissues had peaks. Moreover, the AKP activity in the intestine, with an average of 118.55 ± 23.24 King unit g^−1^ prot, was far greater than that in other tissues.

As shown in Figure 6, the changes in the gill, brain, kidney, and liver tissues were basically consistent, showing a trend of first decreasing, then increasing, and finally decreasing. The changes in the intestine and stomach tissues were basically consistent, showing a trend of first increasing, then decreasing, and finally increasing. The plasma samples showed a trend of first decreasing, then increasing, then decreasing, and finally increasing. The trend of changes in LDH activities in all tissues reached significant levels (*p* < 0.01 or *p* < 0.05). When the light intensity was 18.44–21.04 lx, the LDH activity of most tissues reached a peak, and when the light intensity was 11.15–18.44 lx, the LDH activity of most tissues reached a minimum. LDH activity was relatively higher in the stomach and gill tissues (average value, ≥68.33 ± 17.81 U g^−1^ prot) and lowest in the plasma samples (average value, 3.76 ± 1.03 U g^−1^ prot).

As shown in Figure 7, the changes in the brain, intestine, kidney, and plasma samples were basically consistent, showing a trend of first decreasing, then increasing, and finally decreasing. The change in the liver tissues was basically consistent, showing a trend of first increasing, then decreasing, and finally increasing. The gill and stomach tissues showed a trend of first increasing and then decreasing. Except for brain and stomach tissues, the trend of changes in SOD activity in other tissues reached significant levels (*p* < 0.01 or *p* < 0.05). At light intensity greater than 13.43 ± 2.35 lx, SOD activities were more likely to peak in all tissues. When the light intensity was 11.15 ± 2.01 lx, SOD activities were more likely to be minimal in most tissues. SOD activities were highest in the gill tissues (average value, 30.79 ± 4.11 U mg^−1^ prot) and lowest in the plasma samples (average value, 4.31 ± 0.55 U mg^−1^ prot).

As shown in Figure 8, the changes in the gill, brain, and kidney tissues were basically consistent, showing a trend of first decreasing, then increasing, and finally decreasing. The changes in the stomach and liver tissues were basically consistent, showing a trend of first decreasing, then increasing, then decreasing, and finally increasing, and the trend of changes in the intestine was exactly the opposite. The plasma samples showed a trend of first decreasing and then increasing. The trend of changes in CAT activity in all tissues reached significant levels (*p* < 0.01 or *p* < 0.05). At a light intensity greater than 15.75 ± 2.88 lx, CAT activity was more likely to peak in most tissues. At a light intensity lower than 15.75 ± 2.88 lx, CAT activity was more likely to be minimal in most tissues. CAT activity was highest in the liver tissues (average value, 62.18 ± 16.16 U mg^−1^ prot) and lowest in the plasma samples (average value, 7.04 ± 1.70 U mg^−1^ prot).

As shown in Figure 9, the changes in the brain and kidney tissues were basically consistent, showing a trend of first decreasing, then increasing, and finally decreasing. The plasma samples showed a trend of first increasing, then decreasing, and finally increasing. The changes in the gill, intestine, stomach, and liver tissues were basically consistent, showing a trend of first increasing, then decreasing. The trend of changes in GPT activities in all tissues reached significant levels (*p* < 0.01 or *p* < 0.05). At a light intensity in the range of 13.43–18.44 lx, GPT activity was more likely to peak in most tissues. At a light intensity lower than 13.43 ± 2.35 lx, GPT activity was minimal in most tissues. GPT activity was highest in the stomach tissues (average value, 222.12 ± 84.33 U g^−1^ prot) and lowest in the plasma samples (average value, 27.11 ± 3.13 U g^−1^ prot).

As shown in Figure 10, the changes in the intestine and kidney tissues were basically consistent, showing a trend of first decreasing, then increasing, and finally decreasing. The changes in the liver tissues were basically consistent, showing a trend of first increasing, then decreasing, and finally increasing. The changes in the stomach tissues showed a trend of first increasing, then decreasing, then increasing, and finally decreasing, and the trend of changes in the plasma samples was exactly the opposite. The changes in the gill and brain tissues were basically consistent, showing a trend of first decreasing, then increasing. The trend of changes in GOT activity in all tissues reached significant levels (*p* < 0.01 or *p* < 0.05). At a light intensity lower than 15.75 ± 2.88 lx, GOT activity was more likely to be minimal in all tissues. GOT activity was relatively higher in the stomach, brain, and gill tissues, and the highest activity was in the stomach tissues (average value, 363.41 ± 79.52 U g^−1^ prot). Moreover, the lowest activity was in the plasma samples (average value, 33.16 ± 4.83 U g^−1^ prot), and the activity in the plasma was far lower than that in other tissues.

## 4. Discussion

Satisfactory light affects the growth and reproduction of aquatic species, including *S. chuatsi*, by promoting feeding and the ability to identify potential food sources, especially in the juvenile stage [34,35], which was also reported for the armored catfish (*Sturisoma kneri*) [36]. As opposed to improving the shape, color, flavor, and taste of feed, adjusting the intensity of environmental lighting is more economical and relatively simple. In this study, light intensity altered digestive enzyme activity, demonstrating that appropriate light intensity can increase the activity of digestive enzymes, thereby promoting the digestion ability of food. *S. chuatsi* is a typical carnivorous fish [37] with higher protease and lipase activity than *α*-amylase activity. Gastric tissues play major roles in food digestion of fish [38]. In the present study, the maximum protease and lipase activity in gastric tissues was highest at 18.44 ± 3.00 lx and highest in the intestinal tissues at 11.15 ± 2.01 lx. The maximum protease and lipase activity was higher at 18.44 ± 3.00 lx than at 11.15 ± 2.01 lx, suggesting that gastric tissues may play a greater role in food digestion of *S. chuatsi* and that digestion was strongest at 18.44 ± 3.00 lx, whereas intestinal tissues may play a significant auxiliary role in food digestion of *S. chuatsi* at light intensities of 11.15 ± 2.01 lx. Light colors are also more comforting to some fish [39] and promote feeding but not necessarily growth [15].

To the best of our knowledge, this is the first study to investigate the relationship between the stress response and light in *S. chuatsi*. Although limited, previous studies of light-related stress have mainly focused on cortisol, blood glucose, and other indicators [39,40]. Several of the enzymes selected in this study are usually regarded as indicators of the stress response [41,42]. The gills can be directly exposed to various environmental stimuli, such as hypoxia [43] and nitrite toxicity [44]; thus, the stress response may be stronger than that of other tissues. Accordingly, the results of the present study showed that LDH, SOD, GPT, and GOT activity was relatively higher in gill tissues. The stomach and intestines can also be directly exposed to external stimuli through feeding behavior and food intake, which is a strong inducer of intestinal AKP activities [45]. In this study, ACP, AKP, SOD, and GPT were all highly active in intestinal tissues, whereas the activity of LDH, SOD, GPT, and GOT was relatively higher in gastric tissues. Plasma is the main transporter of molecules throughout the body [46,47]. Hence, various biomarkers can be measured in the plasma of fish. Ryu et al. [48] found differences in plasma levels of SOD and CAT in goldfish (*Carassius auratus*) stimulated at various light intensities, but these differences were not significant. In the present study, ACP was highly active in plasma, and the trend of changes in the other six enzymes also reached a significant level, possibly due to the ferociously carnivorous nature of *S. chuatsi* and a stronger stress response. At a light intensity of 100 – 1600 lx, SOD activity in the liver of Wuchang bream (*Megalobrama amblycephala*) increased, whereas CAT activity first decreased and then increased with increasing light intensity [49], which was contrary to the variation pattern of SOD and was the same as the variation pattern of CAT in the present study, likely due to different species, light intensity, and environmental stimuli.

Recent studies on the response of fish to light have mainly focused on the influence of different colors of light [39,50]. Phototaxis combined with a preference for light colors guide fish to avoid areas that can have negative impacts and prevent gathering in high numbers [10,11,50]. However, relatively few studies have investigated the effect of light intensity on the stress response of fish. Tian et al. [49] found that light intensity at 400 lx was most suitable for the growth of juvenile *M. amblycephala*, and Han et al. [51] found that the intensity of light did not affect the survival rate of *Leiocassis longirostris*, but it significantly affected its growth rate and also affected its body color. Generally speaking, the higher the activity of digestive enzymes, the easier it is for nutrients to be absorbed. Therefore, this study focused on maximum values. At light intensities of 18.44 ± 3.00 lx (protease and lipase in stomach tissues) and 11.15 ± 2.01 lx (protease and lipase in intestine tissues), the maximum value appeared twice. It is worth noting that the activity of the digestive enzymes was greater at 18.44 ± 3.00 lx than 11.15 ± 2.01 lx. In addition, *α*-amylase activity (not the maximum value) was greater at 18.44 ± 3.00 lx than at 11.15 ± 2.01 lx, indicating that the digestive capacity of *S*. *chuatsi* was relatively stronger at 18.44 ± 3.00 lx. Meanwhile, the activity of stress-related enzymes was increased in response to stress. Therefore, the main focus of this study was to determine the light intensity with the greatest occurrence of the minimum value. At a light intensity of 11.15 ± 2.01 lx, the minimum value showed 16 tissues total, which was much greater than with other light intensities. From this, we found that, with the decrease in light intensity, the frequency of the maximum value appeared to decrease, whereas the law of the minimum value was the opposite. From the relevant indicator data of this study, it was found that at a light of 11.15 ± 2.01 lx, the stress response of *S*. *chuatsi* to re-feeding after starvation was relatively low. Previous studies found that the digestive system of *S. chuatsi* can adapt to domestication by feeding an artificial diet [52]. Hence, cultivation of *S*. *chuatsi* under the conditions of 11.15 ± 2.01 lx can facilitate nutrient absorption.

## 5. Conclusions

At 11.15 ± 2.01 lx, *S. chuatsi* sustained relatively lower stress in response to re-feeding after starvation and digestive enzyme activity in the intestine was the highest, indicating that this light intensity is most suitable for re-feeding of *S. chuatsi* after starvation to promote growth and development. Artificial breeding of *S. chuatsi* is rather difficult because of the inclination to prey on live fish. Under light conditions, *S. chuatsi* only preys on fish but will also consume shrimp under dark conditions, and weak light can reduce the accuracy of predation. Thus, under weak light, *S. chuatsi* is most likely to consume compound feed. Moreover, reducing the light intensity increases the activities of digestive enzymes in intestine while alleviating the stress response and will facilitate successful domestication of *S. chuatsi*. Collectively, the results of the present study suggest that a suitable light intensity can accelerate the adaptation of *S. chuatsi* to stress caused by re-feeding and regulate the inclination of *S. chuatsi* to consume compound feed.

## Figures and Tables

**Figure 1 animals-13-02610-f001:**
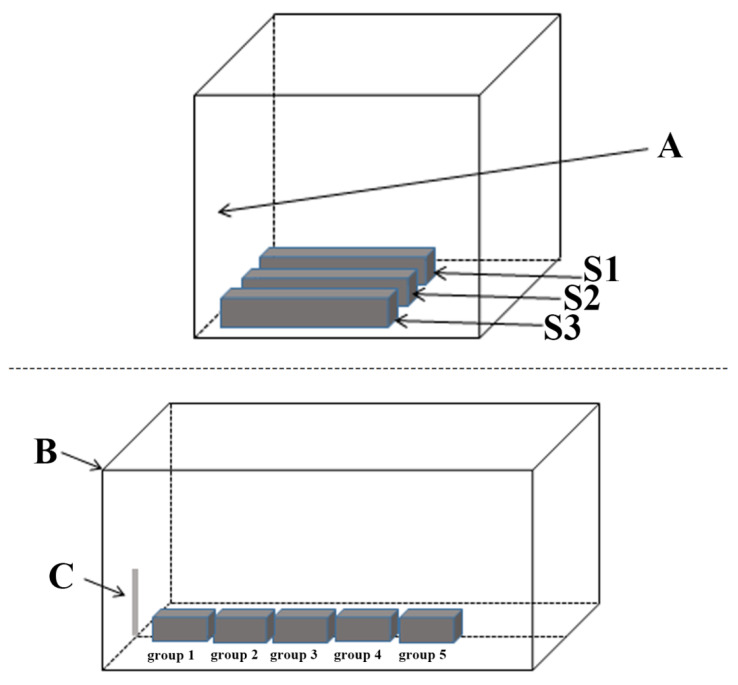
Layout of the experimental facilities. Note: A represents the entire laboratory, where the light source enters from the wall indicated by the arrow. S1, S2, and S3 are identical experimental devices and have a parallel repetition relationship. B is one of S1, S2, and S3, used to demonstrate the experimental setup in detail. C represents the entrance of light, through which natural light can enter the experimental set-up. Groups 1–5 represent the circulating water tanks for the experiment.

**Figure 2 animals-13-02610-f002:**
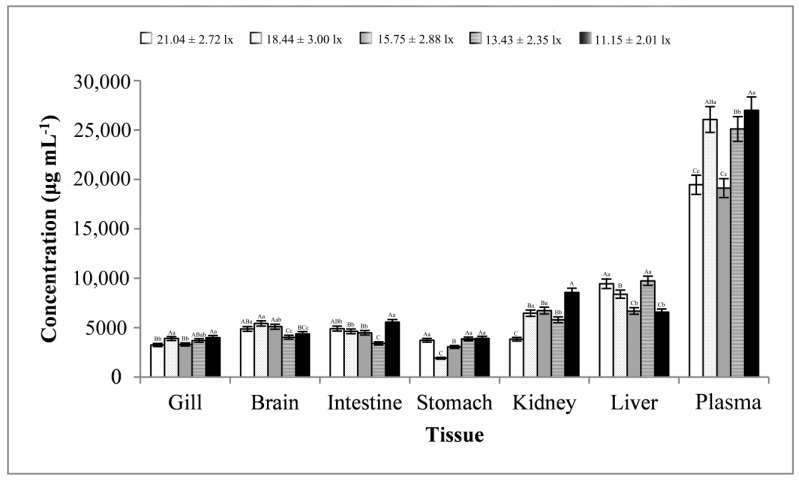
Total protein content of the tissue and plasma samples. Note: The comparison was conducted among different treatment groups within the same tissue. Different capital letters represent extremely significant differences (*p* < 0.01), and different lowercase letters represent significant differences (*p* < 0.05). The same letters mean no significant difference. *n* = 5.

**Figure 3 animals-13-02610-f003:**
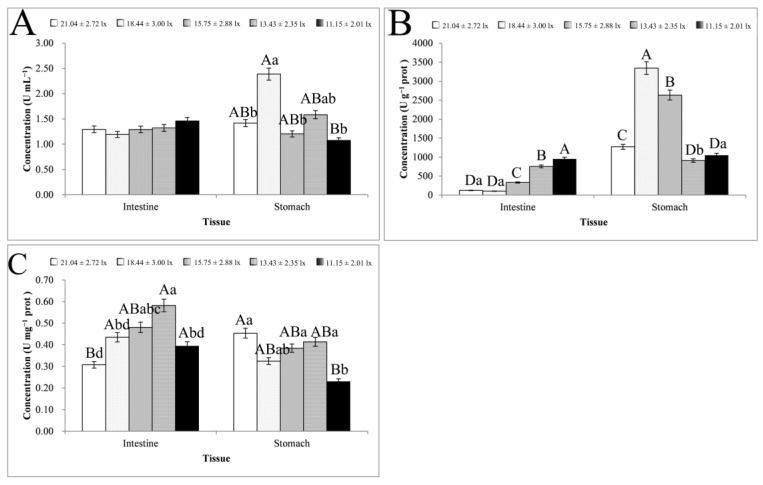
Digestive enzyme activity in the intestine and stomach samples. Note: The letters (**A**–**C**) in the upper left corner of each graph represent protease, lipase, and α-amylase, respectively. The comparison was conducted among different treatment groups within the same tissue. The letters at the top of the bar represent significance. Different capital letters represent extremely significant differences (*p* < 0.01), and different lowercase letters represent significant differences (*p* < 0.05). The same letters or no letters indicate insignificant differences. *n* = 5.

**Figure 4 animals-13-02610-f004:**
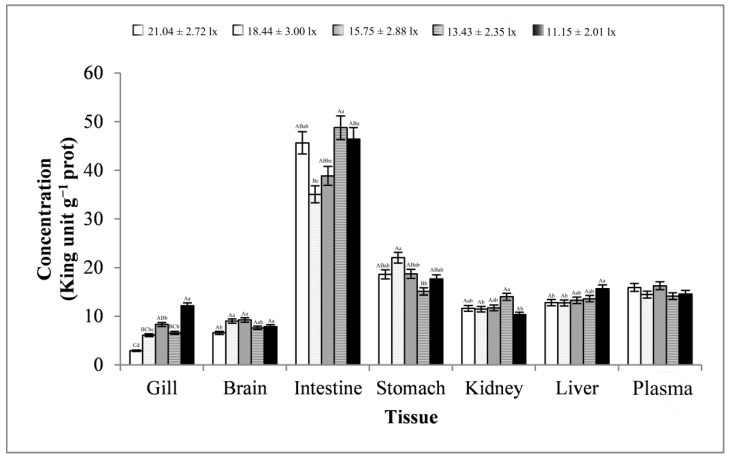
Activity of ACP in different tissues and at different light intensities. Note: The comparison was conducted among different treatment groups within the same tissue. Different capital letters represent extremely significant differences (*p* < 0.01), and different lowercase letters represent significant differences (*p* < 0.05). The same letters or no letters indicate insignificant differences. *n* = 5.

**Figure 5 animals-13-02610-f005:**
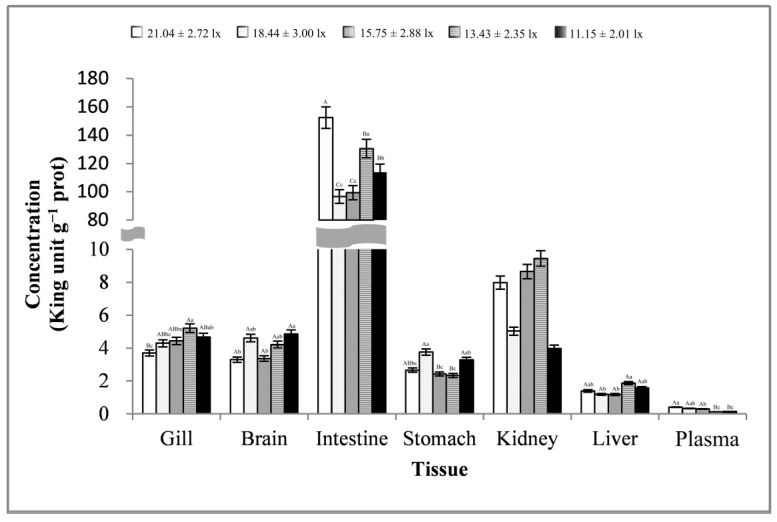
Activity of AKP in different tissues and light intensities. Note: The comparison was conducted among different treatment groups within the same tissue. Different capital letters represent extremely significant differences (*p* < 0.01), and different lowercase letters represent significant differences (*p* < 0.05). The same letters or no letters indicate insignificant differences. *n* = 5.

**Figure 6 animals-13-02610-f006:**
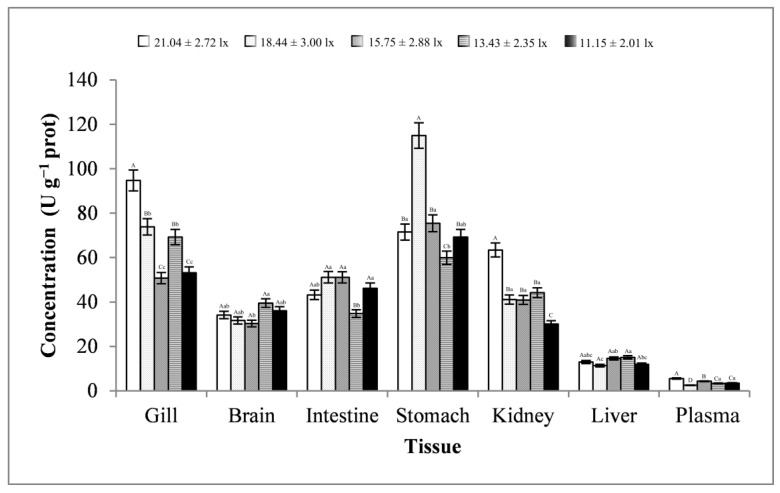
Activity of LDH in different tissues and light intensities. Note: The comparison was conducted among different treatment groups within the same tissue. Different capital letters represent extremely significant differences (*p* < 0.01), and different lowercase letters represent significant differences (*p* < 0.05). The same letters mean no significant difference. *n* = 5.

**Figure 7 animals-13-02610-f007:**
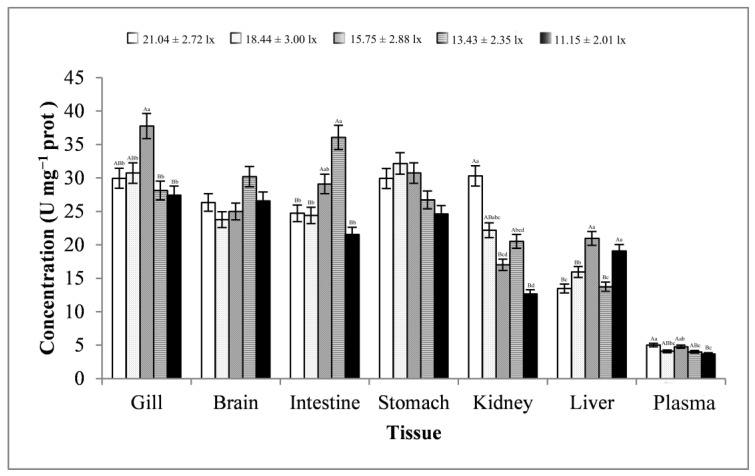
Activity of SOD in different tissues and light intensities. Note: The comparison was conducted among different treatment groups within the same tissue. Different capital letters represent extremely significant differences (*p* < 0.01), and different lowercase letters represent significant differences (*p* < 0.05). The same letters or no letters indicate insignificant differences. *n* = 5.

**Figure 8 animals-13-02610-f008:**
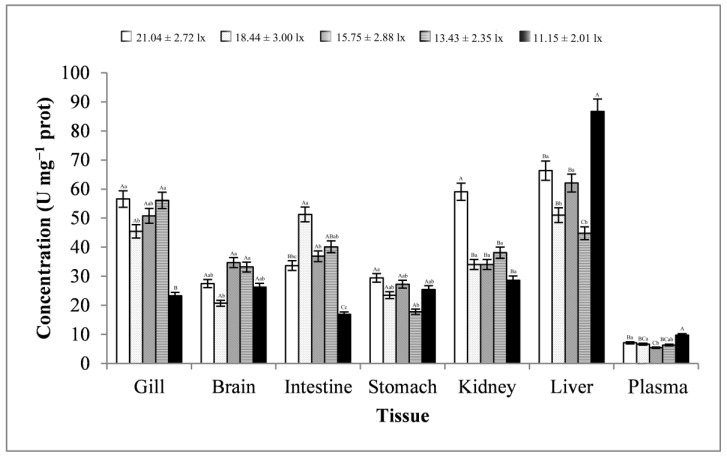
Activity of CAT in different tissues and light intensities. Note: The comparison was conducted among different treatment groups within the same tissue. Different capital letters represent extremely significant differences (*p* < 0.01), and different lowercase letters represent significant differences (*p* < 0.05). The same letters mean no significant difference. *n* = 5.

**Figure 9 animals-13-02610-f009:**
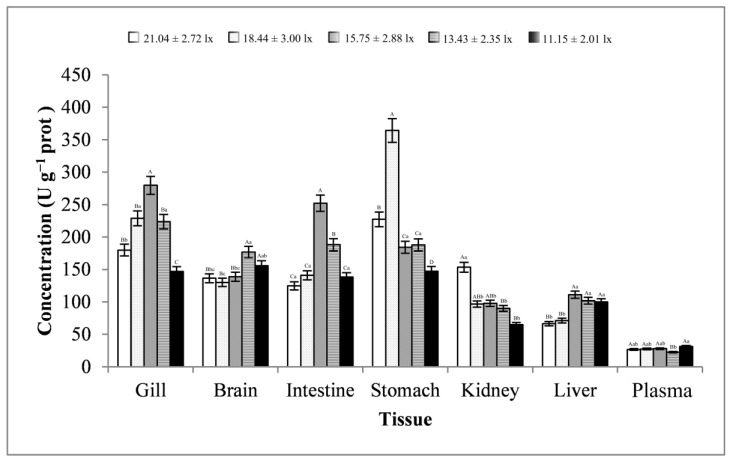
Activity of GPT in different tissues and light intensities. Note: The comparison was conducted among different treatment groups within the same tissue. Different capital letters represent extremely significant differences (*p* < 0.01), and different lowercase letters represent significant differences (*p* < 0.05). The same letters mean no significant difference. *n* = 5.

**Figure 10 animals-13-02610-f010:**
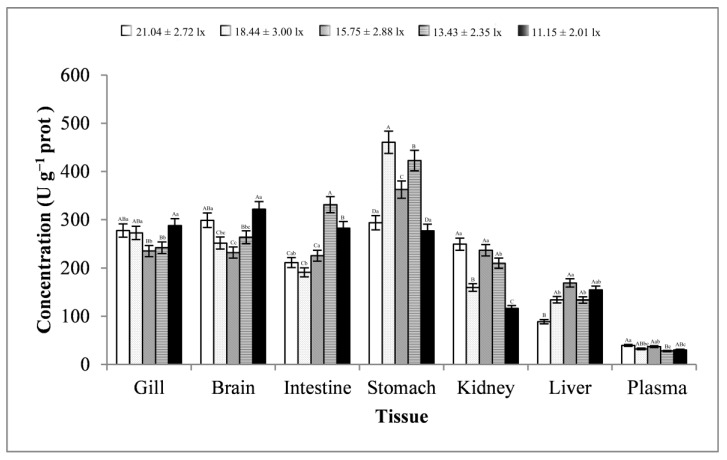
Activity of GOT in different tissues and light intensities. Note: The comparison was conducted among different treatment groups within the same tissue. Different capital letters represent extremely significant differences (*p* < 0.01), and different lowercase letters represent significant differences (*p* < 0.05). The same letters mean no significant difference. *n* = 5.

## Data Availability

All data are available.

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
