# Peer review of "Study on the Adaptive Regulation of Light on the Stress Response of Mandarin Fish (Siniperca chuatsi) with Re-Feeding after Starvation"

_animals, 2023, doi:10.3390/ani13162610_

Round 1

Reviewer 1 Report

For the Study on the adaptive regulation of light on the stress response 2 of Mandarin fish (Siniperca chuatsi) with re-feeding after star-3 vation, the authors studied the changes of digestive enzymes and stress-related enzymes indicators of Mandarin fish (Siniperca chuatsi) under five light intensities21.04 ± 2.72 lx, 18.44 ± 3.00 lx, 15.75 124 ± 2.88 lx, 13.43 ± 2.35 lx, and 11.15 ± 2.01 lx to study the effect of light on the feeding of Siniperca chuatsi. Although there are some problems in the experimental setup and the experimental method is single, the research content of this manuscript has certain novelty and has certain significance for the actual breeding project. It is recommended for Major Revision. The specific comments are as follows:

1There are many grammatical errors in the manuscript, please revise the grammar again.

2Lines 56-79This work did not represent sufficient novelty, as the authors just evaluated the assay of the relevant enzymes in each organ under different light.

3Lines 128-129Why do s1 and s2 choose 2 samples respectively, and s3 chooses 1 sample, what is the reason for the experiment setting up like this? Please explain the reason, and whether it will affect the fairness of sampling and the scientificity of the experimental results.

4Line 192Figure2The text above the histogram is too small to read clearly, and it is unclear what they represent. In addition, please indicate what the group number represents in the legend, not the group 123... or reflect it in the note. For example, 21.04 ± 2.72 lx to replace Group 1 in the legend. Please modify other figures in the manuscript accordingly.

5Lines 110-132Please provide the basis for choosing the light intensity range of 21.04–11.15 lx in the experimental setup in this manuscript.

6Lines 201-202: protease activities in the intestinal tissues under different light are not tended to increase with decreasing light intensity as the author said, how to explain group 5. And there are too many similar inaccurate descriptions in the manuscript of the result trend analysis part. Please make a rigorous description and analyze the trend changes accordingly. Please correct all similar mistakes in the experimental results part.

7Lines 351-375: Too much introduction to the literature review and too little discussion of the mechanisms leading to the findings. Please add relevant description.

8Lines 431-432The author mentioned natural weak light in the morning and evening, or artificial weak light. The experimental design part did not see what kind of weak light source came from, and the experimental results and discussion did not find any description of natural light or artificial light. The author did not do a controlled experiment and cannot prove the consistency of the results of the self-confessed light and artificial weak light experiments. Please describe rigorously according to the experimental settings in the manuscript.

Author Response

For the “Study on the adaptive regulation of light on the stress response 2 of Mandarin fish (Siniperca chuatsi) with re-feeding after star-3 vation”, the authors studied the changes of digestive enzymes and stress-related enzymes indicators of Mandarin fish (Siniperca chuatsi) under five light intensities(21.04 ± 2.72 lx, 18.44 ± 3.00 lx, 15.75 124 ± 2.88 lx, 13.43 ± 2.35 lx, and 11.15 ± 2.01 lx) to study the effect of light on the feeding of Siniperca chuatsi. Although there are some problems in the experimental setup and the experimental method is single, the research content of this manuscript has certain novelty and has certain significance for the actual breeding project. It is recommended for Major Revision. The specific comments are as follows:

1、There are many grammatical errors in the manuscript, please revise the grammar again.

Answer: Thank reviewer for the valuable comments. We corrected it. For example, in Simple Summary, “Environmental factors have significant impacts on the feeding behaviors, yield, and quality of Siniperca chuatsi.” has been changed to “Environmental factors have significant impact on the feeding behavior, yield, and quality of Siniperca chuatsi.” And other grammar corrections would no longer be described one by one.

2、Lines 56-79: This work did not represent sufficient novelty, as the authors just evaluated the assay of the relevant enzymes in each organ under different light.

Answer: Thank reviewer for the valuable comments, and we don't agree with the opinion. Measuring indicators is an important element in measuring the novelty of a study, but more importantly, it should be whether the experimental design of the study is innovative. This study is the first to conduct a correlation study between light and re-feeding after starvation of Mandarin fish based on their living habits and breeding cycles. There is currently no similar research.

3、Lines 128-129,Why do s1 and s2 choose 2 samples respectively, and s3 chooses 1 sample, what is the reason for the experiment setting up like this? Please explain the reason, and whether it will affect the fairness of sampling and the scientificity of the experimental results.

Answer: We apologize for the confusion caused by our mistake. S1, S2, and S3 are identical, and they are parallel controls. The state of the cultured mandarin fish under the same lighting conditions is almost identical, and the number of samples collected will not affect the physiological state of the mandarin fish, so it will not affect the fairness of sampling and the scientificity of the experimental results. The general sampling quantity for a study is at least 3 fish, while in this study, a total of 5 fish were collected under one lighting condition. As there are 3 parallel controls, the design is 2/2/1, and even 3/1/1 will not affect the authenticity and accuracy of the data in this study.

4、Line 192:Figure2,The text above the histogram is too small to read clearly, and it is unclear what they represent. In addition, please indicate what the group number represents in the legend, not the group 1、2、3... or reflect it in the note. For example, 21.04 ± 2.72 lx to replace Group 1 in the legend. Please modify other figures in the manuscript accordingly.

Answer: We apologize for the confusion caused by our mistake. The quality of the images inserted into the manuscript has been compressed, and we have provided the original image (as a standalone file) to clearly read the text inside. And thank reviewer for the valuable comments, we have changed the annotations in the legend.

5、Lines 110-132:Please provide the basis for choosing the light intensity range of 21.04–11.15 lx in the experimental setup in this manuscript.

Answer: We apologize for the confusion caused by our mistake. The light intensity of this study is based on the experience of farmers, and we added relevant information in Source of experimental organism and experimental design (Lines 104-107).

6、Lines 201-202: protease activities in the intestinal tissues under different light are not tended to increase with decreasing light intensity as the author said, how to explain group 5. And there are too many similar inaccurate descriptions in the manuscript of the result trend analysis part. Please make a rigorous description and analyze the trend changes accordingly. Please correct all similar mistakes in the experimental results part.

Answer: Thank reviewer for the valuable comments. We corrected it (Lines 195-200, Lines 284-288, Lines 322-324).

7、Lines 351-375: Too much introduction to the literature review and too little discussion of the mechanisms leading to the findings. Please add relevant description.

Answer: Thank reviewer for the valuable comments. We have streamlined the introduction (Lines 54-68). There are indeed very few studies with high relevance to this study, but after extensive search, a literature (literature 51) was still added in the manuscript.

8、Lines 431-432:The author mentioned natural weak light in the morning and evening, or artificial weak light. The experimental design part did not see what kind of weak light source came from, and the experimental results and discussion did not find any description of natural light or artificial light. The author did not do a controlled experiment and cannot prove the consistency of the results of the self-confessed light and artificial weak light experiments. Please describe rigorously according to the experimental settings in the manuscript.

Answer: Thank reviewer for the valuable comments. We corrected it (Lines 414).

Reviewer 2 Report

The paper investigated the effect of the light intensity on the digestive capacity and stress response of Siniperca chuatsi when re-feeding after starvation. The results would provide some useful references for artificial culture of this commercial species. I recommend it be published after the following issues be further addressed.

1.     The part of “Introduction” is lack of logicality, and many contents here seem no sense for the topic of the paper. I suggest this part be restructured and focus on the paper’s topic, in order to logically state the related background, purpose and significance of this study.

2.     I am confused why the re-feeding was defined as a stress for the S. chuatsi after starvation. Please give some explanations.

3.     Some information of the experimental design should be further clarified or detailed. Please supplement the acclimated conditions of the S. chuatsi in laboratory. It should be clarified how the light intensity were guaranteed in the tanks of the same groups arranged in the three devices (S1, S2, and S3). It should be explained why only one fish individual was sampled from tanks in S3, while 2 ones were sampled from tanks in S1 and S2, respectively.

4.     The changes of each indicator for stress response were depicted respectively, which seems a bit redundant, and too many figures were provided. It is necessary to condense these contents and the figures.

5.     According to the results, the activities of some stress-related enzymes were relatively high even under the weak light (Group 5), and some of them were higher than that under some strong light (Group 1 or Group 2), which is not consistent with some of conclusions in this paper, such as “Moreover, reducing the light intensity increases the activity of digestive enzymes while alleviating the stress response” in Simple Summary, “Collectively, the results showed that light intensity at 11.15±2.01 lx promoted digestive capacity and enhanced anti-stress ability of S. chuatsi in response to stress induced by re-feeding after starvation” in Abstract, and “the stress response of S. chuatsi to re-feeding after starvation was relatively low and more suitable to promote growth” in Discussion. I consider it should be carefully discussed, and the conclusions should be more objective.

Minor issues:

6.     Page 2 Line 52: please confirm the maximum body weight (50kg) of this fish species, and supplement the cited reference here.

7.     Page 2 Line 887: how to define the stress-related enzymes? Are there some references for citing?

8.     Page 3 Line 141: Were the fish euthanized by injection of MS-222? Please supplement the amount of the dose and injection site. Also, if the fish were euthanized by immersing, the concentration of the MS-222 should be supplemented, not the dose.

10. In the notes of Figures 2-10: the tissues were expressed as organization. It is not appropriate.

The language of the text is necessary to be carefully checked and revised.

Author Response

The paper investigated the effect of the light intensity on the digestive capacity and stress response of Siniperca chuatsi when re-feeding after starvation. The results would provide some useful references for artificial culture of this commercial species. I recommend it be published after the following issues be further addressed.

  1. The part of “Introduction” is lack of logicality, and many contents here seem no sense for the topic of the paper. I suggest this part be restructured and focus on the paper’s topic, in order to logically state the related background, purpose and significance of this study.

Answer: Thank reviewer for the valuable comments. We have carefully revised the second paragraph of the introduction to make it more logical (lines 54-75).

  1. I am confused why the re-feeding was defined as a stress for the S. chuatsi after starvation. Please give some explanations.

Answer: We apologize for the confusion caused by our mistake. After experiencing a prolonged period of hunger, fish are in an adaptive state to fasting, and re feeding can produce certain stimuli that may be stronger than normal feeding of fish.

  1. Some information of the experimental design should be further clarified or detailed. Please supplement the acclimated conditions of the S. chuatsi in laboratory. It should be clarified how the light intensity were guaranteed in the tanks of the same groups arranged in the three devices (S1, S2, and S3). It should be explained why only one fish individual was sampled from tanks in S3, while 2 ones were sampled from tanks in S1 and S2, respectively.

Answer: We apologize for the confusion caused by our mistake. For “Please supplement the acclimated conditions of the S. chuatsi in laboratory.”: We added relevant information (lines 99-101). For “It should be clarified how the light intensity were guaranteed in the tanks of the same groups arranged in the three devices (S1, S2, and S3).”: We added relevant information  (lines 107-112). For “It should be explained why only one fish individual was sampled from tanks in S3, while 2 ones were sampled from tanks in S1 and S2, respectively.”: S1, S2, and S3 are identical, and they are parallel controls. The state of the cultured mandarin fish under the same lighting conditions is almost identical, and the number of samples collected will not affect the physiological state of the mandarin fish, so it will not affect the fairness of sampling and the scientificity of the experimental results. The general sampling quantity for a study is at least 3 fish, while in this study, a total of 5 fish were collected under one lighting condition. As there are 3 parallel controls, the design is 2/2/1, and even 3/1/1 will not affect the authenticity and accuracy of the data in this study.

  1. The changes of each indicator for stress response were depicted respectively, which seems a bit redundant, and too many figures were provided. It is necessary to condense these contents and the figures.

Answer: Thank reviewer for the valuable comments. We tried this before, and the sharpness of the pictures was severely reduced when combined, so we had to describe them separately.

  1. According to the results, the activities of some stress-related enzymes were relatively high even under the weak light (Group 5), and some of them were higher than that under some strong light (Group 1 or Group 2), which is not consistent with some of conclusions in this paper, such as “Moreover, reducing the light intensity increases the activity of digestive enzymes while alleviating the stress response” in Simple Summary, “Collectively, the results showed that light intensity at 11.15±2.01 lx promoted digestive capacity and enhanced anti-stress ability of S. chuatsi in response to stress induced by re-feeding after starvation” in Abstract, and “the stress response of S. chuatsi to re-feeding after starvation was relatively low and more suitable to promote growth” in Discussion. I consider it should be carefully discussed, and the conclusions should be more objective.

Answer: Thank reviewer for the valuable comments. We have corrected the issues pointed out by the reviewer to make the expression more accurate (line 20, line 36, line 403).

Minor issues:

  1. Page 2 Line 52: please confirm the maximum body weight (50kg) of this fish species, and supplement the cited reference here.

Answer: We apologize for the confusion caused by our mistake. This data appears on numerous websites, but no official website reports have been found. So I corrected it and introduced new reference (line 50).

  1. Page 2 Line 887: how to define the stress-related enzymes? Are there some references for citing?

Answer: We apologize for the confusion caused by our mistake. In this study, stress-related enzymes refer to enzymes whose expression levels increase or decrease in response to re-feeding after starvation. they have no clear definition, such as Oxidative Stress Related Enzymes (Tsai Y C, Huang C C, Chu L M, et al. Differential influence of propofol on different cell types in terms of the expression of various oxidative stress-related enzymes in an experimental endotoxemia model[J]. Acta Anaesthesiologica Taiwanica, 2012, 50(4): 159-166.).

  1. Page 3 Line 141: Were the fish euthanized by injection of MS-222? Please supplement the amount of the dose and injection site. Also, if the fish were euthanized by immersing, the concentration of the MS-222 should be supplemented, not the dose.

Answer: We apologize for the confusion caused by our mistake. We added the concentration of the MS-222 (line 135).

  1. In the notes of Figures 2-10: the tissues were expressed as organization. It is not appropriate.

Answer: We apologize for the confusion caused by our mistake. We have corrected all the expressions (tissue).

Reviewer 3 Report

In this study, the effect of natural light intensity on digestion capability and stress physiology is investigated in the commercially-important fish species Siniperca chuatsi. Generally, the study seems appropriate to address the questions raised. However, below I have made some suggestions to try and clear away extraneous material that gets in the way of the main message, as well as having some questions about the protocols.

Introduction

1. - L. 74 "Previous studies... ": Can you provide some references here?

2. - Overall, I'm somewhat confused about the point of starvation in this study. It's ostensibly raised as an ecological activity during the winter, but this is in relation to temperature and not light levels. It is often the case that lab animals may be exposed to short periods of starvation to induce feeding, but if that is the only reason for starvation then cut out all the rest. Otherwise it's confusing what the aims of the study are.

3. - Likewise, L. 76, the authors suggest the importance of examining artificial lighting, yet this study focuses (as far as I can tell) on natural lighting. Is it better to just avoid referring to artificial lighting? As it stands I was expecting the authors to explore different light quality as well as intensity.

Methods

4. - thank you for providing information on the ethical approval. Is there a reference for the Guide for the Care and Use of Laboratory Animals?

5. - L. 114 - what were the reduced lighting conditions specifically?

6. - L. 116 - S1 - S3 are somewhat poorly described. I think each could be described as an "aquarium system". Is the interval of 1m between systems (e.g. S1 to S2) or between tanks within system (e.g. S1.1 to S1.2)?

7. - L. 128-130 - why was only 1 fish taken per tank from S3? If there was a limit to how many fish could be further analysed then this should have been randomised. This component of the design has introduced a slight bias.

8. - L. 126 - Feed was withheld for 7 days. Is this biologically relevant e.g. is this a typical fasting time for this fish in the wild? Are the environmental conditions they were held in equivalent to those which induce fasting in the wild? This should be developed in the Introduction. 

9. - L. 141 - How was brain death ensured after overdosing with MS-222?

Discussion

10. - L.391 - not sure it's correct to say that activities of enzymes were a t low levels in brain, kidney and liver. Some were highly expressed in these tissues, and I'm not sure there is anything else you particularly conclude about these tissues, so may be best to remove this sentence.

11. - L. 423 - you have not measured anything to do with breeding, and therefore it is inappropriate to indicate that any of the tested treatment levels will influence breeding.

12. - I find the final paragraph, on maximum and minimum values, both confusing and out of place. I think this needs to be developed in the Results rather than the Discussion, because here you do very little in terms of placing this information within a wider context.

13. - L. 437 - again, you have not measured any effects on feed type, so cannot make this conclusion. However, it would be perfectly fine to indicate the importance of examining light intensity/quality on food preference.

Figures and Tables

14. - Figure 1 - is mg mL^-1 a more appropriate unit for the Y axis?

15. - Figure 2-10 - Group number isn't very informative. Rather than have that at the top, and a key below, why not just have mean lux at the top? The full values are already provided in Table S2, so shortened versions (e.g. 21.0, 18.4, 15.8...) would be fine. The formatting of the letters above bars could be improved - sometimes they are too small, sometimes too big. 

16. - Table S2 - significant differences are exceptionally confusing. Tank 1 has letters Aa, whilst Tank 2 has letters ABb. This means that there is simultaneously a significant difference between tanks (different lowercase letters, a vs b) and no significant difference (A and A). This scheme is too complicated - just have different letters for significant differences.

17. - Do we need data in both a figure and a table? I will leave this decision up to the editor, but typically data are presented in one form, not both, unless there is very good reason.

Author Response

In this study, the effect of natural light intensity on digestion capability and stress physiology is investigated in the commercially-important fish species Siniperca chuatsi. Generally, the study seems appropriate to address the questions raised. However, below I have made some suggestions to try and clear away extraneous material that gets in the way of the main message, as well as having some questions about the protocols.

Introduction

  1. - L. 74 "Previous studies... ": Can you provide some references here?

Answer: We apologize for the confusion caused by our mistake. "Previous studies... " refer to the researches mentioned earlier, and we have changed the description (lines 64-65).

  1. - Overall, I'm somewhat confused about the point of starvation in this study. It's ostensibly raised as an ecological activity during the winter, but this is in relation to temperature and not light levels. It is often the case that lab animals may be exposed to short periods of starvation to induce feeding, but if that is the only reason for starvation then cut out all the rest. Otherwise it's confusing what the aims of the study are.

Answer: We apologize for the confusion caused by our mistake. The core idea of this study is whether appropriate light can alleviate the degree of stress response with re-feeding after starvation. Hunger is only a stimulating factor.

  1. - Likewise, L. 76, the authors suggest the importance of examining artificial lighting, yet this study focuses (as far as I can tell) on natural lighting. Is it better to just avoid referring to artificial lighting? As it stands I was expecting the authors to explore different light quality as well as intensity.

Answer: Thank reviewer for the valuable comments. We have corrected the issues pointed out by the reviewer to make the expression more accurate (lines 65-68).

Methods

  1. - thank you for providing information on the ethical approval. Is there a reference for the Guide for the Care and Use of Laboratory Animals?

Answer: We apologize for the confusion caused by our mistake. This is the animal protection strategy implemented by our unit for experimental animals. There are no relevant reference materials.

  1. - L. 114 - what were the reduced lighting conditions specifically?

Answer: We apologize for the confusion caused by our mistake. A detailed description was provided at Source of experimental organism and experimental design (lines 107-112).

  1. - L. 116 - S1 - S3 are somewhat poorly described. I think each could be described as an "aquarium system". Is the interval of 1m between systems (e.g. S1 to S2) or between tanks within system (e.g. S1.1 to S1.2)?

Answer: We apologize for the confusion caused by our mistake. We corrected it (lines 107-112).

  1. - L. 128-130 - why was only 1 fish taken per tank from S3? If there was a limit to how many fish could be further analysed then this should have been randomised. This component of the design has introduced a slight bias.

Answer: We apologize for the confusion caused by our mistake. S1, S2, and S3 are identical, and they are parallel controls. The state of the cultured mandarin fish under the same lighting conditions is almost identical, and the number of samples collected will not affect the physiological state of the mandarin fish, so it will not affect the fairness of sampling and the scientificity of the experimental results. The general sampling quantity for a study is at least 3 fish, while in this study, a total of 5 fish were collected under one lighting condition. As there are 3 parallel controls, the design is 2/2/1, and even 3/1/1 will not affect the authenticity and accuracy of the data in this study.

  1. - L. 126 - Feed was withheld for 7 days. Is this biologically relevant e.g. is this a typical fasting time for this fish in the wild? Are the environmental conditions they were held in equivalent to those which induce fasting in the wild? This should be developed in the Introduction.

Answer: We apologize for the confusion caused by our mistake. We reviewed a large number of studies on fasting, and 7d is a typical fasting time.

  1. - L. 141 - How was brain death ensured after overdosing with MS-222?

Answer: When it was found that the gill cap was no longer open and closed, it could be confirmed that the fish has entered deep anesthesia.

Discussion

  1. - L.391 - not sure it's correct to say that activities of enzymes were a t low levels in brain, kidney and liver. Some were highly expressed in these tissues, and I'm not sure there is anything else you particularly conclude about these tissues, so may be best to remove this sentence.

Answer: Thank reviewer for the valuable comments. We removed it.

  1. - L. 423 - you have not measured anything to do with breeding, and therefore it is inappropriate to indicate that any of the tested treatment levels will influence breeding.

Answer: Thank reviewer for the valuable comments. We deleted it (line 388).

  1. - I find the final paragraph, on maximum and minimum values, both confusing and out of place. I think this needs to be developed in the Results rather than the Discussion, because here you do very little in terms of placing this information within a wider context.

Answer: Thank reviewer for the valuable comments. We have removed a large number of relevant descriptions and only retained a small portion (lines 396-399).

  1. - L. 437 - again, you have not measured any effects on feed type, so cannot make this conclusion. However, it would be perfectly fine to indicate the importance of examining light intensity/quality on food preference.

Answer: Thank reviewer for the valuable comments. We deleted it (line 406).

Figures and Tables

  1. - Figure 1 - is mg mL^-1 a more appropriate unit for the Y axis?

Answer: Thank reviewer for the valuable comments. When calculating the activities of various enzymes, the calculation is based on the μg/L of protein. If the unit is calculated in mg/L, it may have a significant impact on the results.

  1. - Figure 2-10 - Group number isn't very informative. Rather than have that at the top, and a key below, why not just have mean lux at the top? The full values are already provided in Table S2, so shortened versions (e.g. 21.0, 18.4, 15.8...) would be fine. The formatting of the letters above bars could be improved - sometimes they are too small, sometimes too big.

Answer: Thank reviewer for the valuable comments. We corrected it.

  1. - Table S2 - significant differences are exceptionally confusing. Tank 1 has letters Aa, whilst Tank 2 has letters ABb. This means that there is simultaneously a significant difference between tanks (different lowercase letters, a vs b) and no significant difference (A and A). This scheme is too complicated - just have different letters for significant differences.

Answer: We apologize for the confusion caused by our mistake. This is very necessary. Different capital letters represent extremely significant differences (P < 0.01), and different lowercase letters represent significant differences (P < 0.05).

  1. - Do we need data in both a figure and a table? I will leave this decision up to the editor, but typically data are presented in one form, not both, unless there is very good reason.

Answer: We apologize for the confusion caused by our mistake. We deleted the table.

Round 2

Reviewer 1 Report

This paper is well-revised and can be accepted as it is.

Reviewer 2 Report

The issues I cared were almost addressed. I have no more comments about it. 

The authors didn't represent the language revisions.